# Individualised Exercise Training Enhances Antioxidant Buffering Capacity in Idiopathic Pulmonary Fibrosis

**DOI:** 10.3390/antiox12081645

**Published:** 2023-08-20

**Authors:** Tim J. M. Wallis, Magdalena Minnion, Anna Freeman, Andrew Bates, James M. Otto, Stephen A. Wootton, Sophie V. Fletcher, Michael P. W. Grocott, Martin Feelisch, Mark G. Jones, Sandy Jack

**Affiliations:** 1NIHR Southampton Biomedical Research Centre, Respiratory and Critical Care, University Hospital Southampton, Southampton SO16 6YD, UK; m.minnion@soton.ac.uk (M.M.); a.freeman@soton.ac.uk (A.F.); a.bates@soton.ac.uk (A.B.); james.otto@uhs.nhs.uk (J.M.O.); sophie.fletcher@uhs.nhs.uk (S.V.F.); mike.grocott@soton.ac.uk (M.P.W.G.); m.feelisch@soton.ac.uk (M.F.); mark.jones@soton.ac.uk (M.G.J.); s.jack@soton.ac.uk (S.J.); 2Academic School of Clinical and Experimental Sciences, Faculty of Medicine, University of Southampton, Southampton SO17 1BJ, UK; s.a.wootton@soton.ac.uk; 3Department of Critical Care and Anaesthesia, University Hospital Southampton, Southampton SO16 6YD, UK; 4NIHR Southampton Biomedical Research Centre, Nutrition and Metabolism, University Hospital Southampton, Southampton SO16 6YD, UK; 5Institute for Life Sciences, University of Southampton, Southampton SO17 1BJ, UK

**Keywords:** idiopathic pulmonary fibrosis, exercise training, oxidative stress, redox balance

## Abstract

Exercise training is recommended for patients with idiopathic pulmonary fibrosis (IPF); however, the mechanism(s) underlying its physiological benefits remain unclear. We investigated the effects of an individualised aerobic interval training programme on exercise capacity and redox status in IPF patients. IPF patients were recruited prospectively to an 8-week, twice-weekly cardiopulmonary exercise test (CPET)-derived structured responsive exercise training programme (SRETP). Systemic redox status was assessed pre- and post-CPET at baseline and following SRETP completion. An age- and sex-matched non-IPF control cohort was recruited for baseline comparison only. At baseline, IPF patients (*n* = 15) had evidence of increased oxidative stress compared with the controls as judged by; the plasma reduced/oxidised glutathione ratio (median, control 1856 vs. IPF 736 *p* = 0.046). Eleven IPF patients completed the SRETP (median adherence 88%). Following SRETP completion, there was a significant improvement in exercise capacity assessed via the constant work-rate endurance time (+82%, *p* = 0.003). This was accompanied by an improvement in post-exercise redox status (in favour of antioxidants) assessed via serum total free thiols (median increase, +0.26 μmol/g protein *p* = 0.005) and total glutathione concentration (+0.73 μM *p* = 0.03), as well as a decrease in post-exercise lipid peroxidation products (−1.20 μM *p* = 0.02). Following SRETP completion, post-exercise circulating nitrite concentrations were significantly lower compared with baseline (−0.39 μM *p* = 0.04), suggestive of exercise-induced nitrite utilisation. The SRETP increased both endurance time and systemic antioxidant capacity in IPF patients. The observed reduction in nitrite concentrations provides a mechanistic rationale to investigate nitrite/nitrate supplementation in IPF patients.

## 1. Introduction

Idiopathic pulmonary fibrosis (IPF) is a prototypic progressive fibrotic lung disease [1]. It occurs in middle-aged and elderly adults, with a male predominance (70–80% in an international case series [2,3]) and substantially affects health-related quality of life (HRQoL) [4], with patients typically reporting progressive breathlessness and exercise limitation. Left untreated, the median survival is 3–5 years from the time of diagnosis [5], and there is a clear unmet need for interventions that improve patients’ symptoms.

There is strong evidence that physical inactivity shortens lifespan by increasing the risk of cardiovascular disease and all-cause mortality in the general population [6]. Studies have confirmed that patients with IPF have reduced daily physical activity levels and exercise capacity compared with their peers [7,8]. Increasing exercise intolerance in patients with IPF is associated with worsening quality of life and loss of independence [7,9]. Furthermore, exercise capacity is an independent marker of poor prognosis in patients with IPF with both reduced baseline exercise capacity and declining exercise capacity predictive of increased mortality [10]. Pulmonary rehabilitation, an intervention that includes exercise training, is recommended in international guidelines for patients with IPF, with evidence supporting short-term improvement in functional status [11,12]. Despite this recommendation, it remains uncertain whether standard exercise training provides the optimal strategy for each individual, with up to 50% of IPF patients not improving their 6 min walk test (6 MWT) by greater than the minimum clinically important difference (MCID) of 29–34 m [13,14]. Furthermore, the mechanism(s) underlying any improvement in exercise capacity and whether focused targeting of these could further increase clinical benefit remain uncertain.

Oxidative stress arises due to an imbalance in redox status and regulation, i.e., the relationship between oxidants and antioxidants in the body. Such a redox imbalance, marked by a shift in extracellular redox status in favour of oxidants, has been identified in both bronchoalveolar lavage fluid and blood of patients with IPF compared with aged-matched controls [15,16]. Mechanistic studies have demonstrated that oxidative stress can cause progression of pulmonary fibrosis and markers of oxidative stress correlate with disease severity in IPF patients [17,18]. Whilst acute exercise itself induces oxidative stress [19,20,21], this effect is short-lasting and triggers a hormetic response to increase resilience against oxidative damage. The effects of a period of exercise training on redox status in patients with IPF has not been investigated, whilst evidence from studies in participants without significant respiratory disease suggests that a period of exercise training can alter redox status in favour of antioxidants [22].

Previous studies of exercise training in IPF have used a variety of different exercise protocols [12]. As with pharmacological therapies, the effectiveness of exercise training is dependent on the dose administered [23]. Electro-magnetically braked cycle ergometry combined with an objective assessment of an individual’s physical fitness using cardiopulmonary exercise testing (CPET) enables the prescription of an individualised exercise programme [23]. Using this approach, we have previously demonstrated that an interval-based, structured responsive exercise training programme (SRETP) is feasible and improves physical fitness in patients with cancer [24,25] as well as quality of life and pulmonary inflammation in asthma [26]. Here, we utilise this individualised SRETP approach in a prospective study in patients with IPF to test the hypothesis that SRETP can improve exercise capacity, whilst in parallel investigating its effects on extracellular redox status to provide mechanistic insight into its mode of action.

## 2. Materials and Methods

### 2.1. Study Design

This was a single-centre prospective study. Full ethical approval was obtained (Hampshire A Research Ethics Council, UK [REC: 17/SC/0342]), and the protocol was registered on clinicaltrials.gov (NCT03222648). All patients provided written informed consent before study entry. Study recruitment was between 1 September 2017 and 1 April 2021. However, recruitment and follow-up of IPF patients was terminated on 17 March 2020 due to the COVID-19 pandemic. Hence, only non-IPF participants (controls) were recruited to, and actively participating in, the study from 18 March 2020 to 1 April 2021.

### 2.2. Eligibility

IPF patients aged 18–85 years were eligible to enrol if they had a prior specialist multidisciplinary diagnosis of IPF based on consensus guidelines [27], had a Medical Research Council (MRC) breathlessness score of 1–3, and were clinically stable for 3 months prior to enrolment. Patients were excluded if they were prescribed oxygen therapy, had an FEV_1_/FVC ratio <0.7, had malignancy under treatment/follow-up, participated in pulmonary rehabilitation <6 months prior to screening, or had a condition contraindicating CPET (Appendix A).

An age- and sex-matched control cohort of participants without a diagnosis of IPF were contemporaneously recruited for baseline CPET and redox comparison only. The control participants had been referred for a CPET as part of their standard perioperative care prior to non-malignant elective surgery, having been assessed to have subjectively impaired exercise capacity (Appendix A). The control participants were ineligible for enrolment if they were current smokers, had malignancy under treatment/follow-up, or had a known significant cardiorespiratory disease as judged by the investigator.

### 2.3. Exercise Intervention

All IPF participants attended an in-hospital, supervised 8-week, twice-weekly SRETP. Exercise intensities were determined from symptom-limited CPET (Appendix A) to ensure consistent and individualised exercise for all participants. The exercise intervention was described following the Consensus on Exercise Reporting Template in Appendix A [28]. In brief, exercise training consisted of 40 min interval training on an electromagnetically braked cycle-ergometer that executed intervals automatically. Each interval consisted of 3 min at moderate intensity (80% of work rate at anaerobic threshold (AT)) followed by 2 min at severe intensity (work rate equal to midpoint between AT and peak oxygen consumption ($\overset{\cdot }{V}$O_2_peak)). Training was progressive, with the total number of intervals increasing from 4 to 6 after the first two sessions, and the training intensities were re-prescribed following an interim CPET at week 4. The training programme was adapted from positive studies in patients with cancer [24,25] following patient and carer consultations and expert opinion.

### 2.4. Measurements

#### 2.4.1. Control Participants

Demographics, MRC score and spirometry were recorded, and venous blood sampling (EDTA and serum) for systemic redox profiling was taken immediately pre- and post-CPET. Following this baseline assessment, the control participants had no further active involvement in the study (see Appendix A for the study schedule).

#### 2.4.2. IPF Participants

Demographics, MRC score, spirometry, and gas transfer were recorded at baseline and Week 9. Venous blood (EDTA plasma and serum) for systemic redox profiling was taken immediately pre- and post-CPET at baseline, Week 4, and Week 9.

All CPETs (IPF and control) were analysed by two experienced clinicians (A.F. and A.B.), blinded to patient intervention and outcome. If interpretation varied by ±5%, the CPET was analysed by a third adjudicator (J.O).

### 2.5. Outcomes

In the IPF participants, the study outcomes were assessed at baseline and Week 9. The pre-specified primary outcome was change in endurance time (minutes) at Week 9, assessed using a constant work-rate test with a work rate equal to 75% of baseline $\overset{\cdot }{V}$O_2_peak. The secondary outcomes were (1) CPET variables (AT, $\overset{\cdot }{V}$O_2_peak, and peak work-rate) (2) FVC% predicted, (3) MRC score, (4) St. Georges respiratory questionnaire (SGRQ)—IPF (see Appendix A for a description and Appendix A for the study schedule), and (5) redox biomarkers (see below).

### 2.6. Redox Biomarker Analyses

Whole-body redox status was assessed by measuring the total free thiol concentrations in serum [29] (with higher values reflecting higher antioxidant reserve capacity) and in EDTA plasma based on the ratio of reduced over oxidised glutathione (GSH/GSSG, with lower values representing increased oxidative stress) and low-molecular-weight aminothiol and sulphide concentrations [30]. Whole-body lipid peroxidation was measured in serum using the thiobarbituric acid reactive substance (TBARS) assay [31,32], with 4-hydroxynonenal (4-HNE) protein adducts as complementary markers of lipid oxidation status [33]. Plasma antioxidant capacity was measured using the ferric reducing ability of plasma (FRAP) assay [34]. Nitrite, nitrate, and total nitroso species (RXNO) were assayed in EDTA plasma as biomarkers of nitric oxide (NO) production and metabolism, with cyclic guanosine monophosphate (cGMP) assessed as a marker of downstream NO signalling [35]. For the analytical methodology, see Appendix A.

### 2.7. Statistical Analysis

The data are presented as median and inter-quartile range (IQR), or frequency (*n*) and percentage. Comparisons were made using the Mann–Whitney U test, Wilcoxon signed-rank test, or Chi-squared/Fisher’s exact test as appropriate. Assessments of correlation were made using Spearman’s rank correlation coefficient (r). *p* values < 0.05 were deemed statistically significant. Statistical analysis was conducted using IBM^®^-SPSS^®^ for windows (version 26.0 IBM Corp^®^., Armonk, NY, USA) and Graphpad Prism^®^ (version 9.1 GraphPad Software, Boston, MA USA).

## 3. Results

Fifteen IPF participants were enrolled and completed the baseline assessment, after which one patient withdrew due to an unrelated health condition (Figure 1). The study was halted by the COVID-19 pandemic, precluding ongoing study participation for three patients. Prior to this, 11 patients completed the SRETP with good adherence, median 88% (min. 75%–max. 94%). There were no serious adverse events. Ten control participants were enrolled for CPET blood sampling taken during standard of care perioperative CPET (see Appendix A for indications).

### 3.1. Baseline Demographics

IPF subjects were predominately male (87%) and had a median (IQR) FVC% predicted of 78% (73–88) and TLCO% predicted of 52% (41–66) (Table 1). No significant differences existed for age (median; IPF 72.5 years vs. control 70.4 years *p* = 0.14), sex, or other demographics between the IPF and control participants (Appendix A).

### 3.2. Baseline CPET in Control and IPF Patients

Significant impairment in peak exercise capacity at baseline CPET was apparent in the IPF and control participants, with median predicted $\overset{\cdot }{V}$O_2_peak percent of IPF 66.7% vs. control 64.9% *p* = 0.81 (Appendix A).

The IPF patients exhibited an increased ratio of minute ventilation to volume of carbon dioxide production ($\overset{\cdot }{V}$_E_/$\overset{\cdot }{V}$CO_2_ slope) compared with the controls (indicating increased ventilatory inefficiency), with IPF 41.0 (36–55) vs. control 32.8 (31–36) *p* = 0.002. Exercise-induced desaturation was observed in patients with IPF but not in the controls, with S_p_O_2_ at $\overset{\cdot }{V}$O_2_peak IPF 93% (89–96) vs. Control 99% (98–99) *p* = 0.002.

### 3.3. Systemic Redox Status in Control and IPF Patients at Baseline

The IPF participants had increased basal systemic oxidative stress compared with the controls, evidenced by a significantly lower GSH/GSSG ratio pre-CPET, with IPF 736 vs. control 1856 *p* = 0.046 (Figure 2A). Total GSH levels increased significantly in the controls post-CPET (pre 2.40 μM vs. post 3.25 μM *p* = 0.007), whilst no change was observed in the IPF participants (Figure 2B). No significant differences in either the whole-body redox status assessed via the total free thiol concentrations or whole-body lipid oxidation status assessed via TBARS were observed between the IPF and control participants or from pre- to post-CPET in either group (Figure 2C,D). No significant differences in aminothiols, sulphide, 4-HNE, or FRAP concentrations were observed between the IPF and control participants, either pre- or post-CPET, except for the homocysteine levels, which dropped post-CPET in patients with IPF but not in the controls (Appendix A).

CPET induced a non-significant increase in circulating nitrite concentrations in the controls (pre 0.84 μM vs. post 0.99 μM *p* = 0.155), whilst in the IPF patients, a trend towards a reduction in nitrite was observed post-CPET (pre 1.22 μM vs. post 0.98 μM *p* = 0.133) (Figure 2E). No significant differences in the nitrate (Figure 2F), RXNO (Figure 2G), or cGMP concentrations were observed between the control and IPF participants (Appendix A In the IPF participants, no significant difference was observed in the magnitude of changes in the nitrite or nitrate concentrations pre- to post-CPET when stratifying patients by S_p_O_2_ at $\overset{\cdot }{V}$O_2_peak (<92% vs. ≥92%) ).

### 3.4. Physiological Outcomes following the SRETP in Patients with IPF

Following completion of the 8-week SRETP, the pre-specified primary outcome of the constant work-rate endurance time increased significantly with a median (IQR) improvement of +6.7 min (2.9–14.6), *p* = 0.003 (Figure 3A,B). The median percentage improvement was +82%, with nine participants improving their endurance time by ≥33%, the reported MCID for this variable [36].

The SRETP also led to significant improvements in FVC% predicted (median change +5% *p* = 0.02), peak work-rate (+8 Watts *p* = 0.01), and peak V̇_E_ (+9 L·min^−1^
*p* = 0.03) (Figure 4A–C). No significant changes in participants’ $\overset{\cdot }{V}$O_2_peak, AT, or $\overset{\cdot }{V}$O_2_ work-rate relationship (expressing the oxygen cost of exercise) were observed following the SRETP (Figure 4D–F, Appendix A). Participants’ MRC score improved significantly with *n* = 5, reducing their score by one point (*p* = 0.03). No significant changes in body mass index, $\overset{\cdot }{V}$_E_/$\overset{\cdot }{V}$CO_2_ slope, S_p_O_2_ at $\overset{\cdot }{V}$O_2_peak, or HRQoL were observed following SRETP completion (Table 2, Appendix A).

### 3.5. Effect of the SRETP on Systemic Redox Status

Following the 8-week SRETP, the plasma GSH/GSSG ratio did not change significantly (Figure 5A). However, the SRETP induced a significant increase in post-CPET total GSH concentration, median increase +0.73 μM *p* = 0.028 (+22% from baseline), with total GSH levels increasing significantly pre- to post-CPET at Week 9, with +0.31 μM *p* = 0.021, a response consistent with that observed at baseline in the controls (Figure 5B). Further to this, a significant increase in post-CPET total free thiol concentration was observed following the SRETP, with +0.26 μmol/g protein *p* = 0.005 (+33% from baseline) (Figure 5C).

These increases in post-exercise (CPET) antioxidant reserve capacity were accompanied by a reduction in exercise-induced oxidative stress. Although no significant changes in TBARS were observed post-CPET at baseline and Week 4 (Figure 5D), following SRETP completion (Week 9), TBARS concentration decreased significantly pre- to post-CPET, with −1.20 μM *p* = 0.016 (20% reduction). No significant changes in other aminothiol, sulphide, FRAP, or 4-HNE concentrations were observed following the SRETP (Appendix A).

Following SRETP completion (Week 9), there was a statistically significant decrease in circulating nitrite concentrations pre- to post-CPET (−0.12 μM *p* = 0.013) (Figure 5E). Furthermore, at Week 9 post-CPET, nitrite levels were significantly lower compared to baseline (Week 0), median change −0.39 μM *p* = 0.037 (40% reduction). Nitrate concentrations did not change significantly following the SRETP (Figure 5F). Further, no significant changes in RXNO (Figure 5G) or cGMP concentrations were observed (Appendix A). Stratifying IPF participants by S_p_O_2_ at $\overset{\cdot }{V}$O_2_peak revealed no significant between group difference in the magnitude of the pre- to post-CPET change in nitrite/nitrate concentrations (Appendix A).

## 4. Discussion

We investigated the effect of an individualised aerobic interval training programme on functional capacity and redox status in patients with idiopathic pulmonary fibrosis (IPF). At baseline, IPF patients displayed evidence of increased pre- and post-exercise oxidative stress compared with age-, sex-, and functionally matched controls. The individualised structured responsive exercise training programme (SRETP) was safe and feasible in IPF patients and led to clinically meaningful improvement in endurance time as well as a significant increase in extracellular antioxidant buffering capacity. Our observation that the SRETP progressively potentiated the post-exercise fall in circulating nitrite concentration in IPF patients provides a mechanistic basis to propose that this patient group may benefit from nitrite/nitrate supplementation during exercise training [37].

The recruited control participants provided an appropriately age-, sex-, and functionally matched cohort for robust baseline redox biomarker comparison. Rather than being ‘healthy’ individuals, consistent with the controls representing a cohort of older adults, medical comorbidity was common and further not significantly different to that observed in the IPF group. No significant difference was identified in exercise capacity between participants in the IPF and control groups, an observation consistent with the control patients being prior identified as having subjectively assessed impaired physical fitness necessitating referral for cardiopulmonary exercise testing (CPET) prior to elective surgery. The exercise limitation in IPF participants was primarily due to ventilatory inefficiency and impaired gas exchange, whereas in the controls, it was predominately due to physical deconditioning and cardiovascular limitation.

Reduction and oxidation (redox) processes underpin the architecture of physiological function, and the ability of the body to buffer changes in local and global redox status is essential for the transition between health and disease, and thus resilience [38,39]. It is intrinsically linked to a person’s overall fitness and mitochondrial health, with high oxidative stress loads during exercise proposed to contribute to impaired muscle performance by promoting contractile dysfunction with resultant muscle weakness and fatigue [40]. As patients with IPF appear to have an unfavourable redox status at rest, they are potentially more vulnerable to an oxidative challenge during exercise than age/sex-matched controls [41]. Our finding of a reduced systemic ratio of free glutathione to glutathione disulfide (GSH/GSSG ratio) in IPF patients (an indicator of a shift in extracellular redox state towards oxidation [42]) compared with age-sex matched controls is consistent with the results of Muramatsu et al. [16] and other reports of increased systemic oxidant stress in IPF [15,17,18]. In the present study, we did not observe a significant increase in oxidative stress before (baseline) and immediately after exercise (pre–post CPET) in either the IPF patients or controls. Previous studies into the effect of acute exercise on redox balance in IPF are limited, with Jackson et al. identifying a significant increase in plasma isoprostanes and a reduction in plasma antioxidant capacity [19], whilst similar to our observations, Dowman et al. identified no significant change in oxidant stress assessed via thiobarbituric acid reactive substances (TBARS) [21].

To the best of our knowledge, the effects of a period of exercise training on extracellular redox status in patients with IPF has not previously been investigated. Although we did not observe a change in GSH/GSSG ratio following the SRETP, we identified an increased post-exercise systemic antioxidant buffering capacity using established and validated global assessments of whole-body redox status (increase in plasma total free thiols and total GSH and a reduction in lipid peroxidation products [TBARS]) [30]. The observation of increased antioxidant capacity after exercise training in this study is consistent with previous observations in studies of patients without significant respiratory disease [22], an effect in part thought to be mediated via the upregulation of antioxidant enzymes [43]. Plausibly, if it were shown to be sustained, this increase in extracellular antioxidant buffering capacity could have a prognostic benefit for patients with IPF where oxidative stress is proposed as a key pathway underlying IPF pathogenesis and disease progression [44].

We employed a SRETP with intervals of severe- and moderate-intensity exercise and identified that this approach led to clinically significant improvements in endurance time and peak work-rate. The lack of significant change in patients’ peak volume of oxygen consumption ($\overset{\cdot }{V}$O_2_peak) or anaerobic threshold (AT) is consistent with the previously documented increased sensitivity of constant work-rate tests to therapeutic intervention [45], with improvement in constant work-rate test performance reflecting a change in patients’ ability to sustain submaximal aerobic exercise [46]. Improved submaximal exercise endurance may have a particular physiological benefit for patients with IPF, as it typically reflects the intensity range within which patients with chronic lung disease perform their everyday activities [47]. The improvement in systemic antioxidant capacity we observe identifies a potential mechanism through which muscular fatigue resistance and, consequently, critical power (an individual’s highest obtainable work rate whilst maintaining steady-state physiological conditions) could increase in participants to provide clinical benefit without significant change in aerobic and/or peak exercise capacity, although further investigation would be required to confirm this hypothesis.

We also observed a significant exercise-induced fall in circulating nitrite concentrations in IPF patients upon completion of the SRETP, suggestive of nitrite uptake and/or utilisation. In healthy subjects, transiently increased systemic nitrite concentration post-exercise has been reported by several groups, with the magnitude of this increase positively correlating with individual exercise capacity [48,49]. Traditionally, post-exercise systemic nitrite concentration is thought to represent primarily endogenous NO synthase (eNOS) stimulation in the endothelium of the microcirculation followed by oxidation of the produced NO to nitrite, and reduced bioavailability of endothelial NO is a hallmark of endothelial dysfunction [50,51]. NO produced during exercise acts via several mechanisms to enhance muscle performance including increasing muscle blood flow, improving sarcoplasmic Ca^2+^ handling, and modulating mitochondrial respiratory chain function to improve energy supply/demand [52,53]. The exercise-induced reduction (as opposed to an increase) in circulating nitrite concentration we observed in IPF patients may reflect a compromised endothelial capacity to produce NO with a consequent negative impact on exercise performance. Consistent with this possibility, it has been proposed that there is systemic endothelial dysfunction in IPF patients compared to controls [54,55]. Both aging and senescence induced by telomere shortening have been proposed to induce endothelial dysfunction via oxidative stress [56], suggesting that in IPF patients, any SRETP-induced increase in antioxidant capacity could itself plausibly improve endothelial dysfunction. However, the progressive increase in concentration differences in circulating nitrite pre- to post-CPET throughout the exercise training programme cannot be explained by an existing endothelial dysfunction alone but suggests an increased capacity for utilisation.

An alternative source of nitrite arises from the biotransformation of dietary nitrate in the gastro-intestinal tract [57] and exercising skeletal muscle [58]. In healthy individuals, acute nitrate supplementation has been found to reduce the oxygen cost ($\overset{\cdot }{V}$O_2_) of performing incremental submaximal work on a cycle-ergometer [59]. This observation was contrary to the long-held view that the oxygen cost of performing a sub-maximal work rate is essentially fixed [60]. Although in the present study, there was no indication to suggest improvement in participants’ metabolic/energetic efficiency (based on calculated oxygen cost of exercise), our observation of an exercise-induced fall in nitrite concentration in IPF patients, an effect further potentiated by exercise training, merits further investigation and suggests that IPF patients may benefit from dietary nitrite/nitrate supplementation as an adjunct to exercise training.

This study has several limitations. The sample size is small (and not formally powered to assess positive effect), reflecting the intensive nature of the study, with each patient participating in multiple study visits, enabling investigation of the effect of exercise training on redox status in a comprehensively phenotyped group of patients. No detailed information about patients’ dietary intake and nutritional status has been recorded. A further limitation is the lack of a temporal IPF comparator group who did not undergo SRETP. This however, does not affect the robustness of any conclusions drawn on comparisons before and after direct physiological stress such as CPET. Since the time course of the study was relatively short (9 weeks) and patients were clinically stable for 3 months prior to enrolment, it is unlikely that any change in clinical status impacted our findings. Due to technical considerations of performing CPET, patients requiring oxygen therapy were excluded. Further work is required to translate this method of exercise prescription to this group of patients.

## 5. Summary

In summary, we identified a shift in oxidative tone at baseline in patients with IPF compared with age- and sex-matched controls. We further demonstrated that an individually prescribed aerobic interval training programme is feasible in IPF patients and leads to clinically meaningful improvements in functional status and exercise capacity. Following exercise training, these changes were accompanied by a significant increase in whole-body antioxidant buffering capacity, an effect that may translate into increased resilience against muscle fatigue. We also identified that the exercise programme led to a progressive fall in post-exercise nitrite concentration, an observation suggestive of nitrite utilisation. This was an unexpected finding that provides a mechanistic basis to suggest IPF patients may gain clinical benefit from nitrite/nitrate supplementation during exercise training; this warrants further investigation.

## Figures and Tables

**Figure 1 antioxidants-12-01645-f001:**
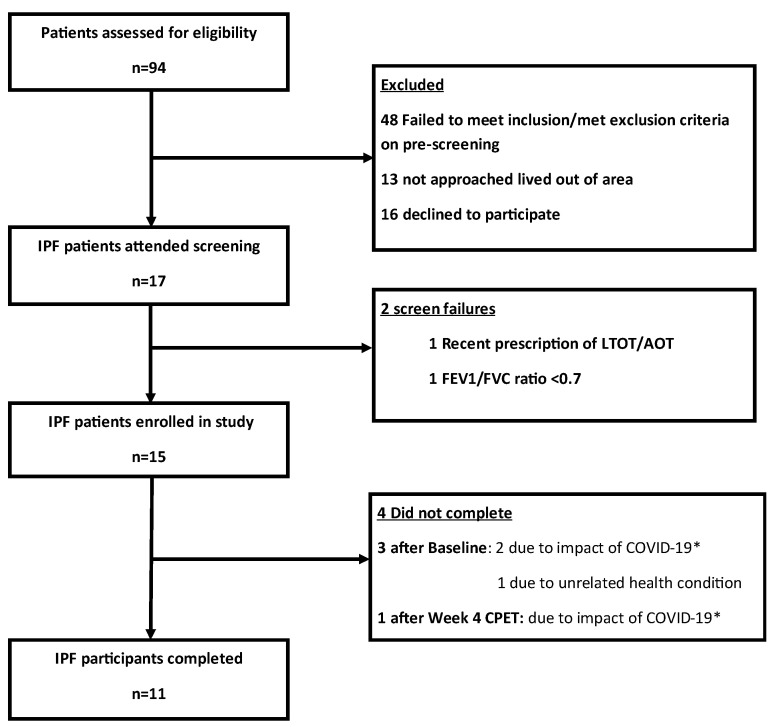
Participant flow diagram. FEV1/FVC—ratio of forced expiratory volume in 1 s to forced vital capacity, LTOT/AOT—long-term oxygen therapy/ambulatory oxygen therapy, * Study recruitment and follow-up assessments of patients with IPF were terminated on 17 March 2020 due to the impact of the COVID-19 pandemic.

**Figure 2 antioxidants-12-01645-f002:**
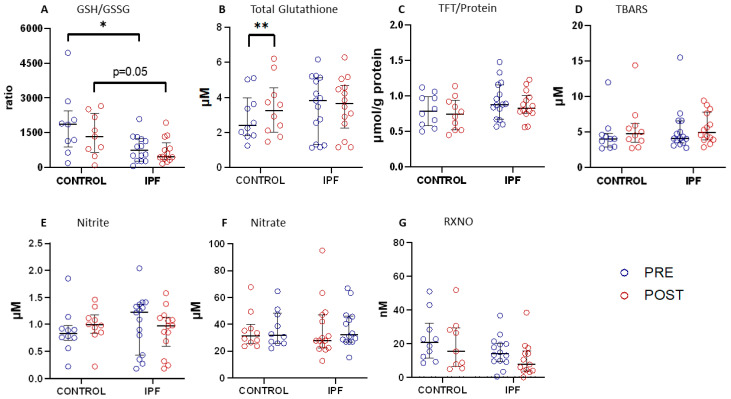
Redox biomarker concentrations at baseline pre- and post-cardiopulmonary exercise testing (CPET) in control vs. IPF participants. PRE-CPET (blue circles) and POST-CPET (red circles). (**A**) GSH/GSSG—ratio of reduced glutathione (GSH) over glutathione disulfide (GSSG)†; (**B**) total glutathione; (**C**) total free thiols (TFT)/protein—total free thiols corrected for protein concentration; (**D**) TBARS—thiobarbituric acid reactive substances; (**E**) nitrite; (**F**) nitrate; (**G**) RXNO (total nitroso species). Control PRE *n* = 10, Control POST *n* = 10, IPF PRE *n* = 15 IPF POST *n* = 14. Error bars median ± interquartile range * *p* < 0.05, ** *p* < 0.01. † Control PRE: GSH/GSSG *n* = 9. IPF PRE: GSH/GSSG *n* = 14 and RXNO *n* = 14. Control POST: GSH/GSSG and RXNO *n* = 9. IPF POST: GSH/GSSG and RXNO *n* = 13.

**Figure 3 antioxidants-12-01645-f003:**
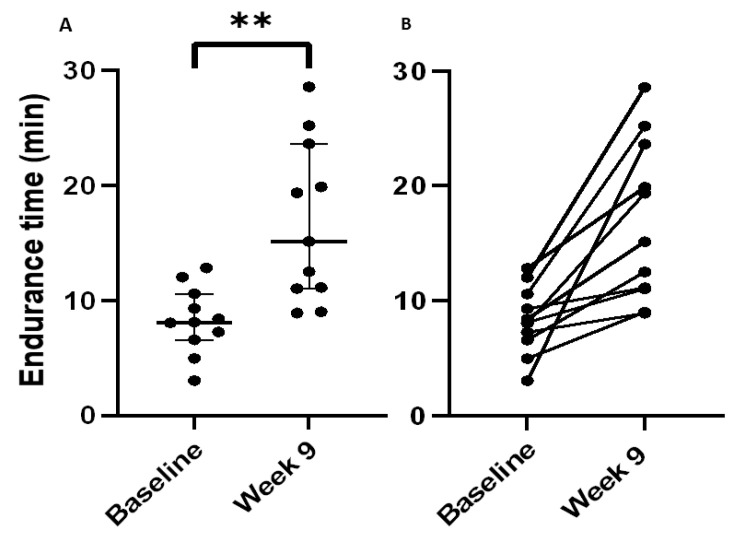
Primary outcome (endurance time) at baseline and Week 9 for IPF patients completing the structured responsive exercise training programme (SRETP). Endurance time (minutes) at baseline and Week 9 on constant work-rate test at 75% of work rate at peak volume of oxygen consumption ($\overset{\cdot }{V}$O_2_peak) for IPF patients completing the exercise programme (*n* = 11). (**A**) Individual dot plot with error bars—median ± interquartile range. (**B**) Line graph of individual patient responses. ** *p* < 0.01.

**Figure 4 antioxidants-12-01645-f004:**
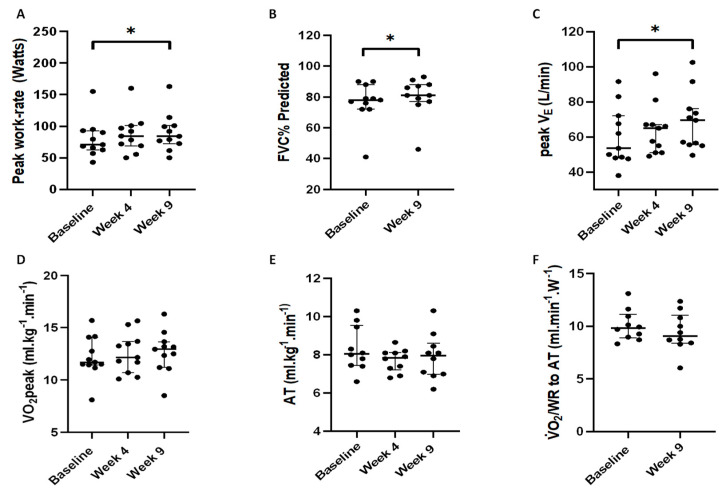
Secondary clinical outcomes for IPF participants completing the structured responsive exercise training programme (SRETP). (**A**) Peak work rate (Watts), (**B**) predicted forced vital capacity percent (FVC% predicted), (**C**) peak minute ventilation (peak $\overset{\cdot }{V}$E, L·min⁻^1^), (**D**) peak volume of oxygen consumption ($\overset{\cdot }{V}$O_2_peak, mL·kg⁻^1^·min⁻^1^), (**E**) anaerobic threshold (AT, mL·kg⁻^1^·min⁻^1^) (**F**) relationship between oxygen uptake and work rate to anaerobic threshold (AT) ($\overset{\cdot }{V}$O_2_/WR to AT) *n* = 11. Error bars - median ± interquartile range.* *p* < 0.05. Note *n* = 10 AT and $\overset{\cdot }{V}$O2/WR to AT.

**Figure 5 antioxidants-12-01645-f005:**
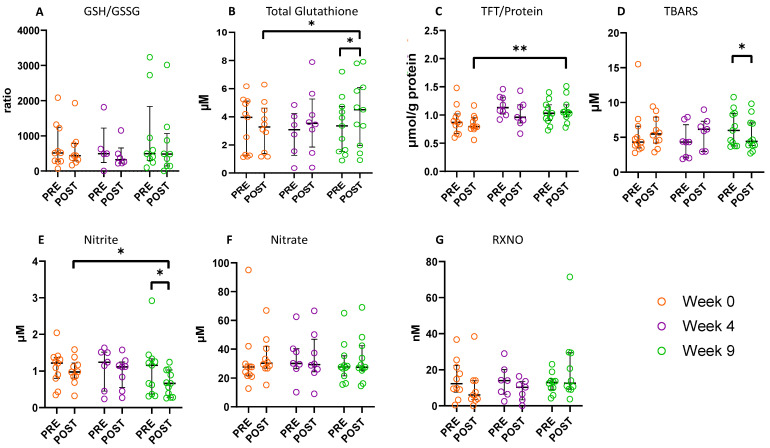
Redox biomarker concentrations pre- and post-cardiopulmonary exercise test (CPET) for IPF patients completing the structured responsive exercise training programme (SRETP). Baseline (orange circles), Week 4 (purple circles), and Week 9 (green circles). (**A**) GSH/GSSG ratio of reduced glutathione (GSH) over glutathione disulfide (GSSG), (**B**) total glutathione, (**C**) total free thiols (TFT)/protein—total free thiols corrected for protein concentration, (**D**) TBARS—thiobarbituric acid reactive substances, (**E**) nitrite, (**F**) nitrate (**G**) RXNO (total nitroso species). Baseline PRE *n* = 11, baseline POST *n* = 10, Week 4 PRE *n* = 8, Week 4 POST *n* = 8, Week 9 PRE *n* = 11, Week 9 POST *n* = 11†. Error bars median ± interquartile range * *p* < 0.05, ** *p* < 0.01. † Baseline PRE: GSH/GSSH ratio and RXNO *n* = 10. Baseline POST: GSH/GSSG and RXNO *n* = 9. Week 4 PRE: GSH/GSSG *n* = 5, total glutathione *n* = 6, nitrite and nitrate *n* = 7, RXNO *n* = 6. Week 4 POST: GSH/GSSG *n* = 6, TBARS, TFT/protein and RXNO *n* = 7. Week 9 PRE: GSH/GSSG *n* = 9 and RXNO *n* = 10. Week 9 POST: GSH/GSSG *n* = 9 and RXNO *n* = 10.

**Table 1 antioxidants-12-01645-t001:** Baseline demographic and exercise capacity in IPF patients recruited to the study.

Demographics (n = 15)	
Age (years)	72.5 (69–80)
Male	87% (13)
BMI (kg·m^−2^)	28.0 (26–32)
FVC% predicted	78 (73–88)
FEV1/FVC	0.82 (0.79–0.84)
TLCO% predicted	52.0 (41.0–62.0)
MRC Breathlessness Score	1–7% (*n* = 1) 2–40% (*n* = 6) 3–53% (*n* = 8)
Antifibrotics	33% (5)
Ex-smoker	67% (10)
Never-smoker	33% (5)
COPD	20% (3)
T2DM	20% (3)
IHD	13% (2)
Baseline exercise capacity (*n* = 15)	
Endurance time (min)	8.05 (5.8–10.6)
$\overset{\cdot }{V}$O_2_peak % predicted	66.7 (57.1–75.9)
Peak WR (Watts)	90 (66–112)
75% $\overset{\cdot }{V}$O_2_peak WR (Watts)	65 (45–85)

Values represented as median (interquartile range) or percentage (*n*). BMI—body mass index, FVC% predicted—predicted forced vital capacity percent, FEV1—forced expiratory volume in 1 s, TLCO% predicted—predicted transfer capacity of the lung for carbon monoxide percent, MRC breathlessness score—Medical Research Council breathlessness score, COPD—chronic obstructive pulmonary disease, T2DM—type 2 diabetes mellitus, IHD—ischaemic heart disease, $\overset{\cdot }{V}$O_2_peak % predicted—predicted peak volume of oxygen consumption expressed as percent, WR—work rate, 75% $\overset{\cdot }{V}$O_2_peak WR—work rate achieved at 75% of $\overset{\cdot }{V}$O_2_peak.

**Table 2 antioxidants-12-01645-t002:** Change in clinical outcomes in IPF patients (*n* = 11) who completed the structured responsive exercise training programme (SRETP).

Variable (Unit)	Median Difference (IQR)	*p* Value
Endurance time (minutes)	+6.7 (2.9–14.6)	0.003 **
FVC% predicted (%)	+5 (1–5)	0.023 *
TLCO% predicted (%)	−2 (−5–+2)	0.121
BMI	−0.26 (−1.25–+1.00)	0.213
AT (mL·kg^−1^·min^−1^)	−0.28 (−1.1–+0.46)	0.484
$\overset{\cdot }{V}$O_2_peak (mL·kg^−1^·min^−1^)	+0.60 (−0.5–+1.2)	0.328
Peak WR (Watts)	+8 (6–16)	0.010 *
$\overset{\cdot }{V}$_E_/$\overset{\cdot }{V}$CO_2_ slope	+2.9 (−6.0–+7.8)	0.286
peak $\overset{\cdot }{V}$_E_ (l·min^−1^)	+9 (2–19)	0.033 *
S_p_O_2_ at $\overset{\cdot }{V}$O_2_peak (%)	−2.0 (−5 to 4)	0.789
SGRQ-I Total Score	−3.9 (−11–+10)	0.790

Values presented as median change (interquartile range). FVC% predicted—predicted forced vital capacity percent. TLCO% predicted—predicted transfer capacity of the lung for carbon monoxide percent, BMI—body mass index, AT—anaerobic threshold, $\overset{\cdot }{V}$O_2_peak—peak volume of oxygen consumption. $\overset{\cdot }{V}$O_2_peak pred—predicted $\overset{\cdot }{V}$O_2_peak expressed as percentage, WR—work rate, $\overset{\cdot }{V}$E/$\overset{\cdot }{V}$CO_2_ slope—slope of relationship between minute ventilation ($\overset{\cdot }{V}$E) and volume of carbon dioxide production ($\overset{\cdot }{V}$CO_2_), SGRQ-I—St. George’s respiratory questionnaire -IPF version; * *p* < 0.05, ** *p* < 0.01. note 10 paired data for AT.

## Data Availability

All data are contained within the article and Appendix A.

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
