# Peer review of "Individualised Exercise Training Enhances Antioxidant Buffering Capacity in Idiopathic Pulmonary Fibrosis"

_antioxidants, 2023, doi:10.3390/antiox12081645_

Round 1

Reviewer 1 Report

This is a very comprehensive reporting of the study describing the usefulness of 8-week structured responsive exercise training program (SRETP). Following 8 wks of SRETP the plasma GSH/GSSG ratio did not change significantly in Idiopathic pulmonary fibrosis (IPF) patients. While, the SRETP induced a significant increase in post-cardiopulmonary exercise test (CPET) total GSH concentration, with total GSH levels increasing significantly pre- to post-CPET at Week 9,  increase in post-CPET total free thiol concentration was observed following the SRETP. Following completion of SRETP, post-exercise circulating nitrite concentrations were found to be significantly lower compared to baseline suggesting exercise-induced nitrite utilization. The SRETP increased both endurance time and systemic antioxidant capacity in IPF patients. The observed reduction in nitrite concentrations could be indicative of an important mechanism to investigate further for nitrite/nitrate supplementation in IPF patients. The conclusion of the study is well-supported by the experimental data. The exercise protocol and the relevant assays for measuring the blood and lung parameters have been adequately described in the supplementary sections. The inclusion and exlusin criteria have been well articulated and necessary institutional approval to conduct the present study has been obtained. Ethical used of human subjects in current research has been appropriately documented.

1.       Why were there an increasing number of male IPF subjects recruited in the study? Was that just by chance or indeed there has been any study showing IPF is more prevalent in males at this specific age group (Avg: 72.5 yrs) that has been reported in the current study?

2.        Due to low number of IPF study participant (enrolled n=15, completed n=11), the scattered data points show heteroscedastic distribution. The authors need to provide additional statistical analysis in order to support that even with such a low number of study participants the reported data is reflective of the trend supporting the study hypothesis. Did the authors consider running a power analysis to determine the smallest number of subjects needed to see a conclusive effect of the exercise training?

Author Response

  1. Why were there an increasing number of male IPF subjects recruited in the study? Was that just by chance or indeed there has been any study showing IPF is more prevalent in males at this specific age group (Avg: 72.5 yrs) that has been reported in the current study?

Thank you for giving us an opportunity to comment on this observation. We believe the sample to be representative of the ratio of males and female patients with IPF in the wider population. IPF is more common in males with international case series identifying a 70-80% male predominance,[1, 2] – This has now been included in the introduction line 50-51.

Moreover, the 2021 UK ILD registry identified that 78.5% of patients diagnosed with IPF in the last 9 years (2013 -2021) recorded on the national registry were male (total IPF cases recorded n=3275).[3] 

  1. Due to low number of IPF study participant (enrolled n=15, completed n=11), the scattered data points show heteroscedastic distribution. The authors need to provide additional statistical analysis in order to support that even with such a low number of study participants the reported data is reflective of the trend supporting the study hypothesis. Did the authors consider running a power analysis to determine the smallest number of subjects needed to see a conclusive effect of the exercise training?

No a priori power calculation was conducted as this study was designed as a pilot study. However, an interim power calculation based on interim data from the first 6 patients completing the SRETP was conducted based on the primary outcome (endurance time). For a continuation of a single-arm study with a baseline endurance time mean of 9.06 minutes ± 3.68, to detect a ≥33% increase from baseline (the proposed MCID for endurance time in patients with COPD) with a Type-I error rate of α=0.05 and 80% power (1-β), required a total sample size of 12 (software used www.ClinCalc.com [accessed 1st August 2019]). A clarification point highlight that the study is not fully powered has been added to discussion limitations section lines 459-450.

Citations

  1. Jo HE, Glaspole I, Grainge C, Goh N, Hopkins PM, Moodley Y, et al. Baseline characteristics of idiopathic pulmonary fibrosis: analysis from the Australian Idiopathic Pulmonary Fibrosis Registry. The European respiratory journal. 2017;49(2).
  2. Fernández-Fabrellas E, Molina-Molina M, Soriano JB, Portal JAR, Ancochea J, Valenzuela C, et al. Demographic and clinical profile of idiopathic pulmonary fibrosis patients in Spain: the SEPAR National Registry. Respiratory research. 2019;20(1):127.
  3. British Thoracic Society ILD Registry Annual Report 2021. Accessed 13th July 2023

Reviewer 2 Report

The study of Wallis and their co-authors is very interesting. The manuscript is well-written and the data is presented clearly. The findings of the study would be very useful for the IPF treatment program. 

There is one minor but important comment. The anamnesis of the patients should include some information about their diet and consumption of supplementary antioxidants, such as nitrate supplements, vitamins,  phytoestrogens, etc. before and during the study. Or, if this information is not available, it should be discussed as a study limitation. There are many in vivo and in vitro studies showing the improvement of pulmonary fibrosis after treatment with different antioxidants. 

Author Response

Response

We thank all the reviewers for their time in reviewing this manuscript and their suggestions and comments which we have reviewed and acted upon and consequently the manuscript has been strengthened and has increased clarity. Please see the response to the individual comments below.

Thank you for this important comment. For further clarification regarding the protocol for the venous blood sampling pre and post cardiopulmonary exercise testing (CPET). All CPETs were conducted in the morning or early afternoon (0800-1300) and patients were instructed to attend the research centre fasted and provided with a light breakfast of toast, butter, spread, jam or cheese at least 2 hours prior to the test. This dietary restriction was necessary to control for nitrogen containing foods which would influence blood analysis of redox markers (personal communication from Professor Martin Feelisch).

One participant was taking the antioxidant N-acetyl cysteine (NAC) at a dose of 600 mg three times a day. This information has now been added to the supplemental file page 3 section 1.1. In addition, we have included an additional sentence to the study limitation section of the main manuscript (line 462-463), as suggested by the reviewer.

Reviewer 3 Report

The authors utilise an individualised exercise training program in a prospective study in patients with pulmonary fibrosis  to test, whether this program can
improve exercise capacity. In parallel, the authors investigate the effects of that training on the extracellular redox status. By doing so, a mechanistic insight into its mode of action shall be provided

A few major and minor comments are listed below

Major

Pls, mention that almost any exercise training improves fitness in almost evrybody and state, why patients suffering from IPF were chosen.

For better readebility, pls introduce subtitles also within the Discussion section

Conclusions. This is a Summary, not a Conclusion. A conclusion would be to employ nitrite/nitrate supplementation for IPF patients. Pls, correct

Minor

p., l. 4 and 5: academic titles are ok? Very unusual.

l. 53: if untreated?

l. 95: intended to last …

l. 187: … was … should read were?

l. 260 & 261: on next page

l. 358 -360: redundant?

l. 388: maybe reference on doi: 10.1007/s40279-014-0149-y

l422: Pls, tell the reader which.

Tab. 2: 6.72 minutes: 2. decimal does not make sense; also with other variables

For the beauty:

Fig. 1: could the boxes have equal width?

Could hyphenation be implemented?

 References

Readability should be improved by indenting the second line

No DOIs?

Author Response

Response

We thank all the reviewers for their time in reviewing this manuscript and their suggestions and comments which we have reviewed and acted upon and consequently the manuscript has been strengthened and has increased clarity. Please see the response to the individual comments below.

Major

Pls, mention that almost any exercise training improves fitness in almost evrybody and state, why patients suffering from IPF were chosen.

We have now included a statement to that matter with an additional reference to the Introduction (lines 56-62) to read as follows:

There is strong evidence that physical inactivity shortens lifespan by increasing the risk of cardiovascular disease and all-cause mortality in the general population [1] Studies have confirmed that patients with IPF have reduced daily physical activity levels and exercise capacity compared to their peers.[2, 3] Increasing exercise intolerance in patients with IPF is associated with worsening quality of life and loss of independence.[2, 4] Furthermore, exercise capacity is an independent marker of poor prognosis in patients with IPF with both reduced baseline exercise capacity and declining exercise capacity predictive of increased mortality.[5]

For better readebility, pls introduce subtitles also within the Discussion section

Thank you for this comment on readability this has been corrected throughout the discussion.

Conclusions. This is a Summary, not a Conclusion. A conclusion would be to employ nitrite/nitrate supplementation for IPF patients. Pls, correct

We have adjusted the heading accordingly.

Minor

p., l. 4 and 5: academic titles are ok? Very unusual.

Agreed - academic titles have been removed from authors’ names.

  1. 53: if untreated?

Yes, this has now been corrected – thanks.

  1. 95: intended to last …

Further sentence added for clarification.

  1. 187: … was … should read were?

This has been corrected thank you

  1. 260 & 261: on next page

Page formatting restructured throughout manuscript.

  1. 358 -360: redundant?

This sentence has been restructured for clarification 336-340.

  1. 388: maybe reference on doi: 10.1007/s40279-014-0149-y

Thank you for the suggestion above reference inserted line 368

l422: Pls, tell the reader which.

This has been inserted in the text line  403-404.

Tab. 2: 6.72 minutes: 2. decimal does not make sense; also with other variables

This has been corrected - thank you.

For the beauty:

Fig. 1: could the boxes have equal width?

This has been corrected.

Could hyphenation be implemented?

Hyphens used as necessary throughout the text

 References

Readability should be improved by indenting the second line

Now reformatted – thanks.

No DOIs?
We have added DOI information to all references cited were they are available.

Citations

  1. Banach M, Lewek J, Surma S, Penson PE, Sahebkar A, Martin SS, et al. The association between daily step count and all-cause and cardiovascular mortality: a meta-analysis. European Journal of Preventive Cardiology. 2023.
  2. Olson AL, Swigris JJ, Belkin A, Hannen L, Yagihashi K, Schenkman M, et al. Physical functional capacity in idiopathic pulmonary fibrosis: performance characteristics of the continuous-scale physical function performance test. Expert Review of Respiratory Medicine. 2015;9(3):361-7.
  3. Wallaert B, Monge E, Le Rouzic O, Wémeau-Stervinou L, Salleron J, Grosbois J-M. Physical Activity in Daily Life of Patients With Fibrotic Idiopathic Interstitial Pneumonia. Chest. 2013;144(5):1652-8.
  4. Chang JA, Curtis JR, Patrick DL, Raghu G. Assessment of health-related quality of life in patients with interstitial lung disease. Chest. 1999;116(5):1175-82.
  5. du Bois RM, Albera C, Bradford WZ, Costabel U, Leff JA, Noble PW, et al. 6-Minute walk distance is an independent predictor of mortality in patients with idiopathic pulmonary fibrosis. The European respiratory journal. 2014;43(5):1421-9.